# *Chirohepevirus* from Bats: Insights into Hepatitis E Virus Diversity and Evolution

**DOI:** 10.3390/v14050905

**Published:** 2022-04-27

**Authors:** Bo Wang, Xing-Lou Yang

**Affiliations:** 1Department of Biomedical Sciences and Pathobiology, Virginia Polytechnic Institute and State University, Blacksburg, VA 24060, USA; 2Kunming Institute of Zoology, Chinese Academy of Sciences, Kunming 650023, China; 3Hubei Jiangxia Lab, Wuhan 430071, China

**Keywords:** hepatitis E virus, *Chirohepevirus*, HEV-D, bats, genetic diversity, genomic catheterization, molecular evolution

## Abstract

Homologs of the human hepatitis E virus (HEV) have been identified in more than a dozen animal species. Some of them have been evidenced to cross species barriers and infect humans. Zoonotic HEV infections cause chronic liver diseases as well as a broad range of extrahepatic manifestations, which increasingly become significant clinical problems. Bats comprise approximately one-fifth of all named mammal species and are unique in their distinct immune response to viral infection. Most importantly, they are natural reservoirs of several highly pathogenic viruses, which have induced severe human diseases. Since the first discovery of HEV-related viruses in bats in 2012, multiple genetically divergent HEV variants have been reported in a total of 12 bat species over the last decade, which markedly expanded the host range of the HEV family and shed light on the evolutionary origin of human HEV. Meanwhile, bat-borne HEV also raised critical public health concerns about its zoonotic potential. Bat HEV strains resemble genomic features but exhibit considerable heterogeneity. Due to the close evolutionary relationships, bat HEV altogether has been recently assigned to an independent genus, *Chirohepevirus*. This review focuses on the current state of bat HEV and provides novel insights into HEV genetic diversity and molecular evolution.

## 1. Introduction

Hepatitis E virus (HEV) is one of the most common causes of acute viral hepatitis. According to the World Health Organization (WHO), there are an estimated 14 million symptomatic cases of hepatitis E annually, with 300,000 deaths and 5200 stillbirths worldwide [1]. HEV primarily transmits to humans through the fecal–oral route via drinking contaminated water in developing countries and through the food-borne route via consuming raw or undercooked animal meats in industrialized nations [2]. HEV usually causes self-limiting acute infection; however, HEV infection in pregnant women shows rapid growth and has a high incidence of developing fulminant hepatic failure with a mortality rate of up to 30% [3]. The altered hormone levels and immunologic responses may contribute to the severity of liver diseases in pregnancy [4]. Furthermore, more than 50% of HEV infections in immunocompromised individuals such as solid organ transplant recipients and patients with HIV infection can progress into chronicity, leading to liver fibrosis, cirrhosis, and death; therefore, chronic HEV infection has recently become a significant clinical problem that requires antiviral intervention [5,6]. Generally, HEV viremia persisting longer than three months can be regarded as a chronic infection and considered for treatment, but treatment options for chronic hepatitis E are very limited since an HEV-specific antiviral is still lacking [7,8]. In addition to fulminant hepatic failure and chronic infection, HEV is also associated with a wide range of extrahepatic manifestations such as neurological and renal injuries [9,10]; however, the underlying molecular mechanisms contributing to the HEV-associated diseases process are poorly understood, and the decisive viral- and host-related factors responsible for relevant clinical disorders remain to be determined.

Hepatitis E was first recognized as enterically transmitted non-A, non-B hepatitis in an outbreak of hepatitis from 1978 to 1979 in the Kashmir valley [11], and shortly thereafter, the causative agent of the disease, HEV, was identified and visualized using immune electron microscopy in late 1981 among Soviet soldiers in Afghanistan [12]. Subsequently, the entire viral genome of the HEV Burma strain from Myanmar was molecularly cloned and sequenced in 1990 [13]. Hepatitis E was initially considered a waterborne disease in endemic areas with poor sanitation until the discovery of genetically homologous HEV strains in pigs in 1997 in the United States [14]. Remarkably, HEV isolated from pigs was shown to infect nonhuman primates, including rhesus macaques and chimpanzees, indicating the possible cross-species transmission and potential zoonotic risk of the virus [6,15]. Soon afterward, sporadic cases of autochthonous hepatitis E were reported in some European countries, particularly in Germany and France, with the consumption of raw and undercooked pork or wild boar products providing direct proof of zoonotic HEV infection [16,17,18,19]. Since the first recognition of chronic HEV infection among liver and kidney transplant recipients with compromised immune systems in 2008 in France, significant concerns about chronic HEV infection in humans have been posed [20]. Although HEV has gained increasing awareness in the past decades, it remains an enigmatic and largely understudied viral pathogen.

HEV is the only one with documented zoonotic transmission among known hepatotropic viruses, and it is estimated to be ranked sixth in spillover potential of 887 wildlife viruses [21]. Since the initial discovery of HEV in pigs (termed swine HEV), a plethora of human HEV genetically close and distant variants was recovered in various animal hosts globally, including avian HEV, rabbit HEV, rat HEV, ferret HEV, bat HEV, camel HEV, and fish HEV [22,23,24,25]. Although the vast majority of zoonotic HEV infection in humans is linked to swine HEV, other variants from deer, rabbits, camels, and rats have also demonstrated their zoonotic potential to cross species barriers and infect humans [16,22,26,27]. The newly discovered HEV-related viruses and their counterpart human HEV have remarkably similar biological properties and share genetic homologies [28]. In general, HEV has a single-stranded, positive-sense RNA genome, which typically encodes three open reading frames (ORFs): ORF1 encodes a nonstructural polyprotein for virus replication; ORF2 encodes the capsid protein; ORF3 partially overlaps ORF2, encodes a multifunctional protein [29,30]. Several putative functional domains have been identified in the HEV ORF1, including methyltransferase (Met), Y domain, papain-like cysteine protease (PCP), hypervariable region (HVR), X domain, helicase (Hel), and RNA-dependent RNA polymerase (RdRp) [31]; however, whether ORF1 polyprotein is cleaved and processed into smaller functional subunits is still debatable [32]. In addition to coding regions, the HEV genome contains a 7-methylguanosine cap (7 mG) at the 5′ untranslated region (UTR) and a polyadenylated (polyA) at the 3′ UTR [33]. Virions of HEV in feces are nonenveloped spherical particles of approximately 27–34 nm in diameter, but virions exist in circulating blood as quasi-enveloped particles with a lipid membrane [34].

Over the past two decades, bats (order Chiroptera) have drawn exceptional attention due to the emergence of severe acute respiratory syndrome coronavirus 1 (SARS-CoV-1), Ebola virus, and recent SARS-CoV-2 since bats are most likely the natural reservoirs of these highly pathogenic viruses [35]. Since the first discovery of HEV-related viruses in worldwide bats in 2012, multiple genetically diversified bat HEV variants have been detected in various Chiropteran species, which further expand the host range of the HEV family [36]. Meanwhile, the newly emerged bat-borne viruses raise concerns about their potential for zoonotic spillover and spread, which might be a threat to global public health [16,37]. In this review, we focus on the current state of bat HEV and anticipate providing new insights into HEV genetic diversity and molecular evolution.

## 2. HEV-D and Bats

HEV is the prototype of the family *Hepeviridae* [23]. According to the 2021 release of the International Committee on the Taxonomy of Viruses (ICTV) (https://talk.ictvonline.org/taxonomy/ (accessed on 2 April 2022)) (Table 1), the family *Hepeviridae* is divided into two subfamilies: *Orthohepevirinae* and *Parahepevirinae*; the former subfamily contains diverse HEV variants from numerous mammals and avians and has been further divided into four genera, *Paslahepevirus*, *Avihepevirus*, *Rocahepevirus*, and *Chirohepevirus*; the latter one contains only fish HEV from cutthroat trout and has a single genus *Piscihepevirus* and species *P. heenan*. The genus *Paslahepevirus* (termed HEV-A hereafter) contains two species, *P. balayani* and *P. alci*, and includes HEV variants primarily from humans, domestic pigs, wild boars, rabbits, deer, and camels [24]. Based on the different evolutionary relationships and host range, eight distinct genotypes thus far have been proposed within the species *P. balayani*: genotypes 1 and 2 HEV (HEV-A1 and -A2) infect humans exclusively in resource-limited countries [38]; HEV-A3 and -A4 cause zoonotic infection in both developing and developed nations, and swine is their main natural reservoir [39]; HEV-A5 and -A6 are discovered in wilds boars in Japan; HEV-A7 is circulating in dromedary camels in Middle East countries with one case of zoonotic HEV-A7 infection reported [40]; HEV-A8 is found in Bactrian camels in China [22]. The species *P. alci* includes an HEV variant from moose (order Artiodactyla). Notably, a single HEV complete genomic sequence from likely tree shrew (order Scandentia) is evolutionarily closely related to known HEV-A variants, whose classification remains unassigned since there is no publication associated with this sequence and its definitive host is unclear. The genus *Avihepevirus* (HEV-B) contains two species, *A. magniiecur* and *A. egretti*, and includes HEV variants from chickens and other avians, including sparrow and little egret [24]. The genus *Rocahepevirus* (HEV-C) contains two species, *R. ratti* and *R. eothenomi*, and includes HEV variants identified in many rodents and carnivores [26]. Lastly, the genus *Chirohepevirus* (HEV-D) contains three species, *C. eptesici*, *C. rhinolophi*, and *C. desmoid*, and includes HEV variants identified in various bats. Nonetheless, multiple genetically distant HEV strains remain unclassified due to the lack of complete genomes and ambiguous phylogenetic positions (Table 1) [41]. Conceivably, the taxonomy of the family *Hepeviridae* continues evolving with the ever-expanding host range of HEV [42].

Bats are the second most abundant vertebrates after rodents. To date, 21 bat families, over 200 genera, and more than 1400 species have been discovered on this planet, which comprises approximately 20% of all extant mammal species [43]. Uniquely, bats are the only flying mammals with a long lifespan compared with other placental mammals of similar body sizes [44]. Furthermore, certain bat species have specific biological features such as hibernation, echolocation, and crowded roosting [45]. Of utmost importance, bats play essential roles in ecosystems since they disperse seeds, pollinate plants, and consume tons of agriculturally harmful insects [46]. Over the past two decades, bats have drawn unprecedented attention because they are recognized as potential reservoirs for several highly pathogenic emerging viruses, which have induced severe diseases and outbreaks in humans and livestock [35,45]. For instance, in the 1990s, the spillover of Hendra and Nipah viruses originated from fruit bats (likely *Pteropus* spp., flying foxes) have caused outbreaks in Australia and Malaysia, respectively [47]; during 2002–2003, the SARS-CoV-1 originated from insectivorous bats (likely *Rhinolophus sinicus*, Chinese horseshoe bat) was responsible for more than 8000 cases of human infections globally with a mortality rate close to 10% [48]; during 2014–2016, the Ebola virus caused devastating outbreaks in west Africa with mortality rates reaching 39.5%, and several fruit bat species have been suggested as its reservoirs [49]. More recently, as of March 2022, the ongoing Coronavirus Disease 2019 (COVID-19) pandemic caused nearly 490 million human infections and more than 6 million deaths around the world (https://coronavirus.jhu.edu/map.html (accessed on 18 March 2022)), the causative agent of this emerging catastrophic disease, SARS-CoV-2, has been linked to some horseshoe bat species [50]. In addition, a large number of previously unknown viral pathogens have been detected in bat specimens, with the application and advancement of high-throughput sequencing techniques in metagenomic and bat virome studies [51,52]; however, why bats harbor many viruses with significant genetic heterogeneity and how bats live harmoniously with some highly lethal viruses are still elusive [35]. One plausible hypothesis is that an inhibitory immune state may exist in bats and the tolerance of viral infection leads to asymptomatic infection of bats, thus continuous viral replication and virus shedding in bats [53].

The first study of the HEV-D detection in bats was published in 2012; in total, 3869 fecal, liver, and blood specimens involving 85 bat species from nine countries on four continents were screened for HEV and HEV-related viruses RNA. In consequence, HEV-D has been discovered in five bat species from three countries, which are genetically close relatives of human HEV-A and form a monophyletic clade within the HEV family [36]. Of note, the first and representative full-length genome of HEV-D from a German *Eptesicus serotinus* bat (termed BS7/EptSer) has been sequenced and extensively characterized in this milestone study (Table 2). Subsequently, HEV-D was identified in a Chinese myotis bat and three Japanese microbats, with one sequenced viral genome from a *Myotis davidii* bat (Md2350/MyoDav) [54,55]. Additionally, diverse HEV-D sequencing reads and contigs, including one assembled complete genome from a *Pipistrellus nathusii* bat (Ps1/PipNat), four from *Desmodus rotundus* bats (AYA11/DesRot, AYA11/DesRot, API17/DesRot, LR3/DesRot), one from a *Rhinolophus ferrumequinum* bat (SX2013/RhiFer), have been found in various bat species from four metagenomic analyses targeting bat virome, which were conducted by different research groups in the world [56,57,58,59]. Collectively, these studies strongly indicate that bats are the natural reservoirs of genetically diversified HEV-D variants; however, the knowledge of biology, ecology, pathogenesis, evolution, and interspecies transmission of HEV-D with bat origin are scarce and merit further investigation.

Overall, during the last decade, partial and complete genomic sequences of bat HEV-D have by far been detected in five families of 12 Chiropteran species in eight countries, including Germany, Japan, China, Switzerland, Panama, Peru, New Zealand, and Ghana (Table 2). For a better presentation, worldwide detection and distribution of bat HEV-D are depicted in Figure 1. Given the diverse and geographically dispersed bat species on Earth and the growing focus on bat-borne emerging infectious diseases, as well as zoonotic HEV infections, it is unarguable that increasing bat HEV-D variants will be disclosed in the future.

## 3. Genetic Diversity of Bat HEV-D

A total of 19 bat-derived HEV-D sequences are available in the Database of Bat-Associated Viruses (DBatVir) (http://www.mgc.ac.cn/DBatVir/ (accessed on 2 April 2022)), which derived from 18 bat HEV-D strains since one short subfragment is just the Sanger sequencing confirmation of a metagenomic contig [56]. Among them, eight viral genomes have been obtained from five Chiropteran species, which provide an excellent opportunity for a comparative genomic analysis to delineate the intra- and inter-genera/species genetic diversity of the genus *Chirohepevirus* within the family *Hepeviridae*. The sequence comparisons of five designated HEV genera demonstrate that bat HEV-D has very low homologies with the other three HEV genera (HEV-A to -C) (Figure 2a), with identities less than 50% at both nucleotide and amino acid levels, let alone the fish HEV within the sister subfamily, with identities of only 33.3% at the nucleotide level and 22.4% at the amino acid level. Of note, the identities are comparably similar (about 45%) between each of the four *Orthohepevirinae* genera but more divergent (~33% at nucleotide, ~22% at amino acid) from a single *Parahepevirinae* genus, indicating that the current taxonomy of the family *Hepeviridae* is reasonable. Additionally, sequence analysis based on the pairwise distance of respective four genera (HEV-A to -D) shows highly similar maximum values (around 0.4) at both nucleotide and amino acid levels within each of the genera (Figure 2b), which further indicates the rationality of the present division of the subfamily *Orthohepevirinae* into four distinct genera [60]. Nonetheless, it is noteworthy that the HEV variant from the likely tree shrew has been recognized as HEV-A for sequence comparisons and pairwise distances analyses; otherwise, there will be a significantly narrower pairwise distance of HEV-A [60]. Likewise, HEV strains from wild field mice and voles were included in HEV-C as suggested previously [61]. On the contrary, several unassigned HEV strains from wild hamsters were excluded from HEV-C due to their controversial phylogenetic placement [26].

As outlined above, according to the latest suggestion of the ICTV *Hepeviridae* Study Group, the genus *Chirohepevirus* has been further divided into three species, *C. eptesici*, *C. rhinolophi*, and *C. desmoid*. The species *C. eptesici* includes two HEV-D genomes derived from *E. serotinus* and *M. davidii* bats; the species *C. rhinolophi* includes a single HEV-D genome derived from *R. ferrumequinum* bat; the species *C. desmoid* includes four HEV-D genomes derived from *D. rotundus* bats. However, an HEV-D genome derived from *P. nathusii* bat is not yet assigned, likely because the very recent discovery of the virus falls behind the classification [59]. Sequence comparisons of complete genomes between eight bat HEV-D strains show that the homology is highly correlating with the host phylogeny (Figure 2c), such as bat families and species. Specifically, HEV-D from different bat families has very low sequence identities of approximately 60% at both nucleotide and amino acid levels. By contrast, the BS7/EptSer, Md2350/MyoDav, and Ps1/PipNat from the same bat family *Vespertilionidae* share relatively higher sequence identities of 67.2–70.8% at nucleotide and 73.4–78.6% at amino acid, indicating that the Ps1/PipNat could be included in the species *C. eptesici*. Moreover, the AYA11/DesRot, AYA11/DesRot, API17/DesRot, LR3/DesRot from the same bat species *D. rotundus*, exhibit even higher sequence identities of 66.0–94.9% at nucleotide and 72.2–98.8% at amino acid, despite the striking divergence of the LR3/DesRot with the other three DesRot HEV-D variants [58]; therefore, the genomic sequence analyses highly support the current species diversification of the genus *Chirohepevirus*. Additionally, it is hypothesized that HEV-D and their Chiropteran hosts have likely undertaken the virus–host co-speciation [28]. Similarly, the amino acid identity plot of concatenated ORF1 and ORF2 reveals that all bat HEV-D strains have a similar identity tendency over the viral genome compared with the human HEV-A (Figure 2d). As anticipated, the highest identities (nearly 90%) between bat HEV-D and human HEV-A occur at three defined functional domains, including Met, Hel, and RdRp, which is in agreement with the bioinformatic analysis from previous studies and likely associated with the conservation of structural conformation of HEV ORF1 [31,36].

Based on the comprehensive phylogenetic analysis and specific host tropism, the assignment of HEV genotypes has also been assigned for HEV-A (A1 to A8) and HEV-C (C1 to C3) (Table 1) [41,60]; however, the definitive demarcation criteria of genotype assignment are inconsistent with regard to distinct HEV genera or species since the pairwise distances between genotypes of HEV-C are obviously much broader than that of the HEV-A [60]. Considering the remarkable genetic variability and distance of bat HEV-D, multiple genotypes could also be assigned with viruses from respective host families and/or species, for instance, HEV-D strains from *D. rotundus* bats. Nonetheless, given the different geographical distribution and significant epidemiological relevance of genotypes A1 to A8, genotype differentiation for HEV-A has enormous biological and clinical soundness [22,24,62]. Similar to zoonotic HEV infections induced mostly by HEV-A3 and -A4 but not HEV-A1 and -A2, only HEV-C1 derived from rats (also possibly intermediate host, shrews) has so far been evidenced to infect humans, whereas HEV-C2 and -C3 is unlikely zoonotic spillover to humans or nonhuman primates [63]; therefore, the necessity of genotype assignment for HEV-D remains to be determined.

**Figure 2 viruses-14-00905-f002:**
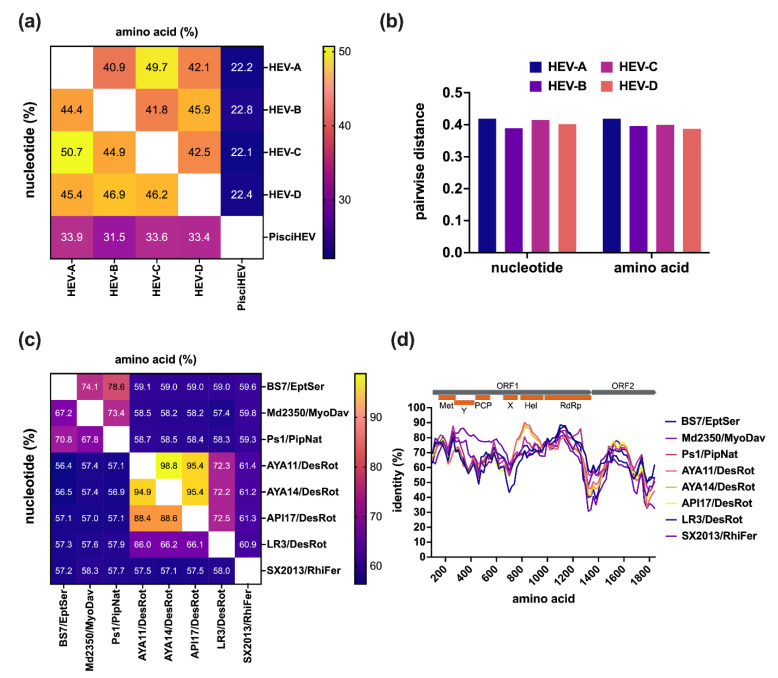
Intra- and inter-genus/species comparative analysis of bat HEV. (**a**) Comparisons of nucleotide of complete genomes and amino acids of concatenated ORF1 and ORF2 between five distinct HEV genera. (**b**) Pairwise of maximal distances per each of four HEV genera within the subfamily *Orthohepeviridae* of amino acid of concatenated ORF1 and ORF2 and nucleotide of complete genomes. (**c**) Comparisons of nucleotide of complete genomes and amino acid of concatenated ORF1 and ORF2 and between eight bat HEV strains. (**d**) Amino acid identity plot of concatenated ORF1 and ORF2 of bat HEV strains to the HEV prototype Burma strain (GenBank accession no. M73218). A schematic representation of concatenated ORF1 and ORF2 with HEV putative functional domains (Met, Y, PCP, X, Hel, RdRp) is shown at the top. Amino acid and nucleotide sequences were aligned using the MAFFT algorithm in Geneious Prime version 2022.0.2. Evolutionary analyses are conducted in MEGA version 11.0.11 [64]. Numbers in each of the cells represent the average identity percent between the two groups. The heatmap is generated using Prism version 9.3.1. Identity plot was performed in SimPlot version 3.5.1 with a window size of 200 and a step size of 20 amino acids increment [65]. Positions containing gaps and HVR regions were stripped from the alignment. GenBank accession numbers of representative HEV strains of each genus used for analyses: M73218 (HEV-A1), KX578717 (HEV-A2), AF082843 (HEV-A3), AB197673 (HEV-A4), AB573435 (HEV-A5), AB602441 (HEV-A6), KJ496143 (HEV-A7), KX387865 (HEV-A8), KF951328 (HEV-A from moose), KR905549 (HEV-A from likely tree shrew), AY535004 (HEV-B form chicken), KX589065 (HEV-B from little egret), GU345042 (HEV-C1 from rat), JN998606 (HEV-C2 from ferret), MG020022 (HEV-C from field mouse), MG020024 (HEV-C from vole), JQ001749 (BS7/EptSer), KX513953 (Md2350/MyoDav), MT815970 (Ps1/PipNat), MW249012 (AYA11/DesRot), MW249013 (AYA14/DesRot), MW249011 (API17/DesRot), MW249014 (LR3/DesRot), KJ562187 (SX2013/RhiFer), HQ731075 (pisciHEV from fish).

## 4. Genomic Characterization of Bat HEV-D

Bat HEV-D and other mammalian and avian HEV variants resemble the genomic structure and ORF composition, despite the significant genetic divergence in sequence as described above [28]. Generally, bat HEV-D variants have comparably shorter genome lengths, from ~6.5 kb (SX2013/RhiFer) to ~6.8 kb (BS7/EptSer), compared with human HEV strains of ~7.2 kb; however, it is intriguing that they have markedly longer ORF3 (138–150 aa) than human HEV ORF3 (113 aa) in size. In fact, the ORF3 in bat HEV-D tends to be the longest of all mammalian and avian HEV variants so far discovered (Figure 3). Moreover, ORF3 from bat HEV-D is entirely located inside the N-terminal of ORF2, in contrast to other mammalian and avian HEV variants whose ORF3 overlaps partially with the N-terminal of ORF2. Additionally, the identities of ORF3 sequences between each HEV-D variant from different bat species are up to approximately 70%, but less than 20% compared to other mammalian and avian HEV strains. Recently, it has been well-characterized that ORF3 acts as a functional viroporin and facilitates the release of infectious virions from infected cells [66]. The high conservation of ORF3 from respective animal origin and its low similarity from different hosts strongly imply that ORF3 may also play a critical role in the HEV host tropism.

The major functional domains within ORF1 can be found in all bat HEV-D strains (Figure 4a), supporting the existence and conservation of Met, Hel, and RdRp domains in HEV genomes from different animal origins [31,36]. Furthermore, specific motifs in each of three functional domains identified in human HEV-A from previous studies are also highly conserved between eight HEV-D strains derived from different bat hosts (Figure 4b–d), which is probably associated with the maintenance of their secondary structure and enzyme functionality, and thus explains the highest identities in the three functional regions of bat HEV-D genomic identity plot as shown in Figure 2d [31]. Although several other putative domains, including the Y domain, PCP, HVR, X domain, with unknown functions in HEV nonstructural polyprotein can also be traced in bat HEV-D genomes, they are much less conserved [36].

In addition to the three typical ORFs in the HEV genome, novel ORFs have recently been identified in specific HEV genotypes and strains (Figure 3). For example, a novel viral factor (designated ORF4), located in HEV ORF1, has been identified in HEV-A1 but not in other HEV-A genotypes. This unique ORF4 protein is induced by endoplasmic reticulum (ER) stress, interacts with several host and viral proteins, and assembles a protein complex stimulating the HEV RdRp activity [67]. Experimental evidence demonstrated the ectopic expression of the ORF4 could enhance HEV-A3 replication in cultured human hepatoma cells [68]. Another novel ORF (tentatively named ORF4), located at the 5′ end of HEV ORF1, is ubiquitous in rat HEV-C1 and ferret HEV-C2 strains but is not found in HEV-C variants from other wild rodents [26,27]. The functional role of this specific ORF4 remains to be determined since a recent study has shown that it is not essential for the in vitro active replication of rat HEV [69]. Interestingly, several putative ORFs, with extremely low homology to any defined domain of functional proteins, have been predicted in some bat HEV-D variants [36]; however, the experimental confirmation of their functional roles is still lacking.

## 5. Molecular Evolution of Bat HEV-D

Historically, the mammalian order Chiroptera used to be divided into two suborders, Megachiroptera and Microchiroptera, based on the distinct bat morphology; however, with the advancement of molecular approaches and the discovery of ancient fossil records, the order has been firmly re-differentiated into Yangochiroptera and Yinpterochiroptera suborders, which is estimated to diverge around 63 million years ago by molecular clock analysis [70,71]. Most importantly, recent studies have predicted that the order Chiroptera host a high proportion of zoonotic viruses and significant viral diversity relative to other mammalian orders [72], which is reflective of the substantial number of bat species [73]. Over the past decade, sufficient evidence from independent studies has confirmed that bats also harbor HEV-related viruses, which are genetically distant from other mammalian and avian HEV variants and form an independent monophyletic clade within the HEV family [36,54,55,56,57,58,59].

In order to systematically delineate the evolutionary history and phylogenetic relationship of bat HEV-D, we compiled HEV-D sequences so far available in the GenBank, which contain eight complete genomes and ten partial genomic sequences as described above; however, the partial short HEV-D sequences are located at different regions of the viral genome due to the application of PCR assays aiming at different conserved viral genomic regions or the metagenomic sequencing methods producing viral reads and contigs randomly. Mapping partial sequences to the reference HEV Burma strain shows that a single HEV-D sequence comprising 217 amino acids locates overlapping Hel and RdRp regions (Figure 5a). In comparison, seven HEV-D sequences containing 108 amino acids locate inside the RdRp region [36,56]. In addition, three HEV-D sequences comprise nearly the entire RdRp and capsid gene [55]; therefore, we conducted two separate phylogenetic analyses with partial bat HEV-D sequences ranging from 217 amino acids (includes ten HEV-D sequences) and 108 amino acids (includes 17 HEV-D sequences) in HEV ORF1, respectively (Figure 5b,c).

As illustrated in Figure 5b, HEV strains cluster together within their respective viral genus (HEV-A to -D), apart from one virus from New Zealand *Mystacina tuberculate* bat, which amazingly segregates all other mammalian and avian HEV variants [56]. The placement of this HEV-related virus from *M. tuberculate* bat strongly suggests it can constitute a novel genus, if not a subfamily, within the HEV family. Notably, the Chiropteran species *M. tuberculate* is supposed to be isolated from other terrestrial mammals on the Zealandia subcontinent for over 16 million years, which probably leads to the viral–host co-evolution and partially explains the considerable divergence between its specific HEV variant and other mammalian and avian HEV variants; however, it is still noticeable that the viral contig (673 bp) is detected in bat guano samples and has the highest amino acid sequence identity of 30.3% to the fish HEV isolated from a cutthroat trout; thus, a spillover infection from other aquatic animal species cannot be excluded. The availability of the complete genome of HEV-related virus from *M. tuberculate* bat guano samples will be beneficial in identifying its precise animal origin and definitive taxonomical position.

Similarly, in Figure 5c, all HEV-D sequences from a total of 4 bat families and 11 species in the world are monophyletic within the family *Hepeviridae*, which is consistent with the findings in previous studies, further indicating that HEV-D variants are evolutionarily linked to their bat hosts and supporting the virus–host co-speciation hypothesis [36,55,58]. Notably, the substantial genetic diversity of four bat HEV-D variants from *D. rotundus*, as well as the apparent longer branch length of HEV-D in the phylogeny compared to that of the HEV-A, strongly imply a long-term HEV-D evolution and adaptation in each bat host. Nonetheless, it is also not neglectable that the placement of HEV-B significantly differs from each other in phylogenetic trees using two distinct partial ORF1 regions, suggesting that phylogeny based on short viral genomic sequences should be interpreted cautiously, especially for those highly distant viruses.

Moreover, to provide a clearer insight into the molecular evolution of bat HEV-D, we performed complete genomic phylogeny of HEV-D variants in relation to their chiropteran hosts (Figure 6a–e). As shown in Figure 6a, the topologies of phylogenies of the complete viral genome and ORF1 Hel and RdRp 217 amino acids are highly comparable, but the complete viral genome phylogeny has significantly higher posterior probabilities at each node, implying greater certainty and accuracy in the phylogenetic reconstructions. As expected, bat-borne HEV-D represents a separate evolutionary lineage within the *Hepeviridae* family, which is evolutionarily distant from HEV variants from other vertebrate hosts. Notably, bat HEV-D strains are further grouped into various sub-clades within the genus *Chirohepevirus* (Figure 6a), which provides valuable information to support investigations of the evolution of bat-originated HEV-D.

It is anticipated that an increasing number of genetically diverse HEV-D will be identified in various bat species in the future since the HEV-D has thus far only been discovered in five Chiropteran families and 12 species (Figure 6b), which is barely 1% of a total of 1435 named Chiropteran species. In the host phylogeny of bats harboring HEV-D (Figure 6c), 12 species are clearly grouped into their respective lineages of five families. Furthermore, the Chiropteran suborder Yangochiroptera (contains bat families *Vespertilionidae*, *Phyllostomidae*, and *Mystacinidae*) and the other suborder Yinpterochiroptera (contains *Rhinolophidae* and *Hipposideridae*) also significantly diverge with each other in the phylogenetic tree, which is consistent with the current classification of bat subordinal relationships based on previous phylogenomic analysis [71]. Notably, the complete viral genome phylogeny of HEV-D exhibits a highly similar phyletic clustering with that of their corresponding bat families and species (Figure 6d). Specifically, although HEV-D strains from *M. davidii* and *R. herrumequinum* bats have been discovered in China, they are phylogenetically divergent due to their distinct host families. By contrast, HEV-D strains from *M. davidii* bat in China, *E. serotinus* bat in Germany, and *E. serotinus* bat in Switzerland, within the same bat family *Vespertilionidae*, share a common ancestor irrespective of their different continental habitations; therefore, a combination of virus and host phylogenies strongly emphasizes the virus–host co-speciation bat HEV-D.

However, the phylogeny of partial ORF1 (327 nt) comprising more HEV-D sequences seems highly complex (Figure 6e). It is surprising that the HEV-D strain from *M. davidii* bat is basal to all other HEV-D strains from its own bat family *Vespertilionidae* and those even from distant bat families *Phyllostomidae* and *Hipposideridae* in the phylogenetic tree. Moreover, HEV-D variants from Yinpterochiropteran *H. abae* bat cluster with HEV-D variants from other Yangochiropteran bats, which conflicts with bat evolution. Nonetheless, as discussed earlier, it should be noted that phylogeny based on partial short viral sequences could not always correctly project the evolutionary history of HEV. Hence, whether interspecies transmission and genetic recombination events occur in the history of bat HEV-D evolution remains unknown.

In fact, the evolutionary patterns are incredibly intricate with respect to the family *Hepeviridae*. The phylogenetic relationships of HEV are not always consistent with that of vertebrate orders. For instance, rats (order Rodentia) and rabbits (order Lagomorpha) are evolutionarily closely related and jointly belong to the grandorder Glires; however, rat HEV-C1 and rabbit HEV-A3r are evolutionarily highly divergent, especially rabbit HEV-A3r clusters into HEV-A3 originated from humans (order Primates) and diverse even-toed ungulates (order Artiodactyla) [74]. Recently, it has been hypothesized that human HEV-A might be introduced from ancestor viruses of HEV-C, which has a significantly greater degree of viral and host diversity [26,27,75]; subsequently, ancient HEV-A was accidentally cross-species transmitted to some even-toed ungulates as well as rabbits, likely due to the intensive husbandry farming [76]; ultimately, present HEV-A1 to -A4 have undergone genotype-specific evolution with the different viral fitness and eventually result in their distinct host range, geographical distribution, and infection pattern [77]. Nonetheless, there are still many questions that remain unexplored in HEV evolution and diversification. The amplification and identification of complete genomes of genetically heterogeneous HEV-related viruses in other mammalian orders will undoubtedly provide a better understanding of the evolutionary history and origin of human HEV.

## 6. Infection Patterns of Bat HEV-D

Our current knowledge of biological characteristics and infection patterns of bat HEV-D is still very limited since the majority of bat HEV-D studies are restricted to viral surveillance in field-collected bat specimens. Nonetheless, the discovery of diverse bat HEV-D variants raises significant public health concerns regarding its zoonotic potential, considering that bats are natural reservoirs of several emerging zoonotic viruses, and multiple HEV variants from different animal species have demonstrated their zoonotic risks [6,22,35]. Given that all the bat HEV-D strains constitute an independent monophyletic clade within the family *Hepeviridae*, which is highly divergent from the phyletic clade of human HEV-A strains, it is thence unlikely that bat HEV-D can cross species barriers to infect humans. Furthermore, a total of 93,146 plasma samples from blood donors in Germany and 453 serum samples from HIV-infected patients in Cameroon were tested negative for bat HEV-D RNA, indicating that there is no evidence of zoonotic infection of bat HEV-D [36]; however, the possibility of zoonotic transmission of bat HEV-D cannot be wholly ignored since the divergent rat HEV-C1, which was previously thought restricted to rats and shrews, has recently been reported to infect humans in a series of cases in Hongkong, Canada, and Spain from independent studies [63,78].

Despite the remarkable genetic divergence, bat-borne HEV-D has been revealed to share similarities in infection patterns with HEV-A [28]. For example, high amounts of viral RNA have been observed in the liver specimens of a German *E. serotinus* bat and a Chinese *M. davidii* bat, suggesting an apparent hepatotropism of HEV-D [36,54]. Notably, viral RNA has also been found in more than a dozen tissue specimens, including the bat kidney and brain, which strongly indicates that extrahepatic viral replication may also occur concerning HEV-D [36,54]; however, whether HEV-D causes extrahepatic manifestations and injuries in bats is unclear. In view of the relatively low detection rates of HEV-D, it is not likely that there is a long-standing viral shedding [36]; however, the low detection rates of HEV-D in bats may be associated with primers mismatch of viral family-based consensus PCR methods, which can underestimate the infection status of bat HEV-D to a great extent [36,54,55]. Indeed, a majority of genetically distant HEV-D sequences are derived from deep sequencing methods unbiasedly targeting bat virome [56,57,58,59]. Taken into account that plentiful studies have indicated that bats are more tolerant of viral persistence due to their unique natures of immune defenses [35,45,79], and chronic HEV infections have so far been documented for human HEV-A3, rabbit HEV-A3r, and ferret HEV-C2 [22]. Whether HEV-D also induces prolonged or even chronic viral infections in bats is still to be determined. Most importantly, the HEV-D associated disease pathology in bats is yet largely unknown.

To date, fundamental research in terms of molecular virology and mechanisms of virus–host interactions for the HEV-D has been poorly studied due to the absence of useful virological tools such as reverse genetics and cell culture systems [80]. Furthermore, the in-depth investigation of viral pathogenicity of HEV-D requires a suitable animal model. Presumably, bats can serve as a naturally occurring animal model for HEV-D, but it is technically challenging to work with these animals, and the immunology and immunopathology of bats are not fully understood [35,81]. Nevertheless, further comparative studies of functional and structural characterization of bat HEV-D will provide implications for human HEV replication and pathogenesis.

## 7. Conclusions

Hepatitis E is an emerging but largely underdiagnosed disease. Its causative agent HEV is distinctive in that it is the only zoonotic hepatotropic viral pathogen with numerous animal reservoirs. Zoonotic viral spillover has already been documented for diversified homologs of human HEV derived from swine, camels, rabbits, and even rats. Importantly, zoonotic HEV infection leads to chronic liver diseases and a broad spectrum of extrahepatic manifestations in humans, which has recently become a significant clinical problem. With the unprecedented focus on global bat virome and the development and advancement of sophisticated sequencing techniques, accumulating genetically divergent HEV-related viruses have been identified in various bat populations during the past decade, which poses critical public health concerns of their potential risk of cross-species transmission to humans. Taxonomically, bat HEV strains have been assigned to an independent genus *Chirohepevirus* (termed HEV-D) within the family *Hepeviridae*. Despite the significant phylogenetic distance from human HEV, the possibility of a bat-to-human spillover of HEV-D cannot be completely neglected. Notably, the discovery of bat-borne HEV-D has remarkably expanded the host range and genetic diversity of the HEV family and will shine new light on the natural history and evolutionary origin of human HEV. Moreover, further detection and genomic characterization of increasing HEV-D variants in other bat species will be advantageous to accurately dissect the molecular evolution of HEV-D, practically to identify the possible interspecies transmission and genetic recombination events. Future functional research is required to elucidate the virus ecology and molecular biology of bat HEV-D.

## Figures and Tables

**Figure 1 viruses-14-00905-f001:**
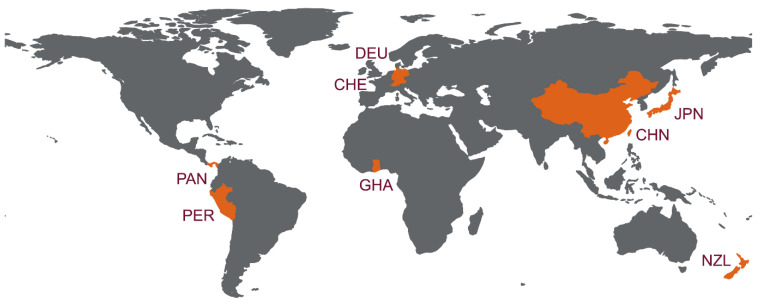
Worldwide distribution of HEV-D in bats. Countries with HEV-D genomic sequence detected are indicated in orange. Three-digit codes for each of the countries: CHE: Switzerland; CHN: China; DEU: Germany; GHA: Ghana; JPN: Japan; NZL: New Zealand; PER: Peru; PAN, Panama. The world map is created using a free and open-source quantum geographic information system (QGIS) version 3.22 (https://qgis.org/ (accessed on 18 March 2022)) and free vector and raster map data from Natural Earth (https://www.naturalearthdata.com/ (accessed on 18 March 2022)).

**Figure 3 viruses-14-00905-f003:**
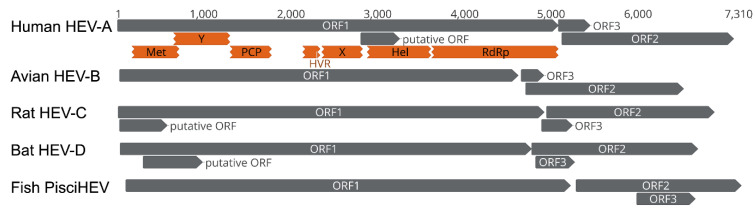
Comparisons of the genomic organization of human HEV-A, avian HEV-B, rat HEV-C, bat HEV-D, and fish PisciHEV. The genome scale in nucleotide is shown on the top. Three principal HEV ORFs and additional putative HEV ORFs, as well as putative functional domains within the ORF1 of human HEV-A, are depicted. The putative ORF in human HEV-A only exists in the majority of HEV-A1 strains. GenBank accession numbers of representative HEV strains of each genus: M73218 (human HEV-A), AY535004 (avian HEV-B), GU345042 (rat HEV-C), JQ001749 (bat HEV-D), HQ731075 (Fish PischHEV).

**Figure 4 viruses-14-00905-f004:**
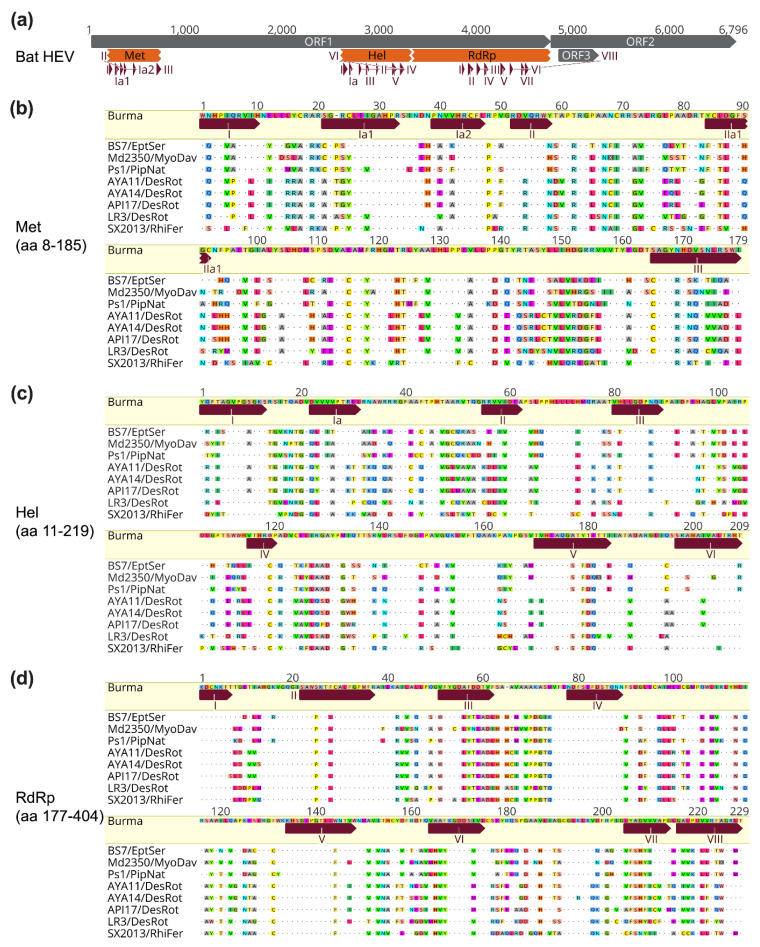
Conservation of bat HEV functional domains and motifs. (**a**) A schematic diagram of the genomic organization of bat HEV (JQ001749) and its ORFs, functional domains, and motifs are shown. (**b**–**d**) Amino acid sequence alignments of reported motifs in three functional domains of Met, Hel, RdRp of bat HEV strains with the Burma strain (M73218). Positions of HEV functional domains and motifs are according to the Burma strain and are based on [31,36]. Complete genomes and functional domains are aligned using the MAFFT algorithm in Geneious Prime version 2022.0.2. Amino acid residues identical or different to the reference are in dots or colors, respectively. GenBank accession numbers of bat HEV-D strains used for analyses: JQ001749 (BS7/EptSer), KX513953 (Md2350/MyoDav), MT815970 (Ps1/PipNat), MW249012 (AYA11/DesRot), MW249013 (AYA14/DesRot), MW249011 (API17/DesRot), MW249014 (LR3/DesRot), KJ562187 (SX2013/RhiFer).

**Figure 5 viruses-14-00905-f005:**
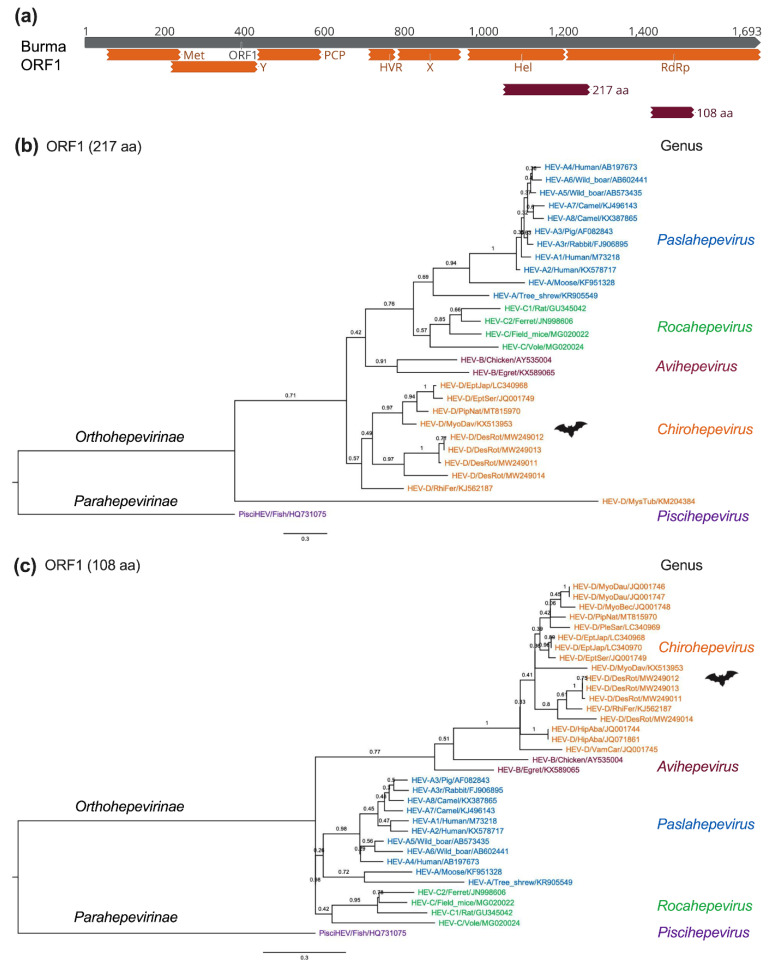
Phylogenetic relationships of bat HEV within the *Hepeviridae* family. (**a**) Bat HEV partial ORF1 amino acid sequences were mapped to the ORF1 of the HEV prototype Burma strain (GenBank accession no. M73218). The positions of 217 aa (1049 to 1265) and 108 aa (1419 to 1526) are shown and used for sequence alignments and phylogenetic analyses. (**b**,**c**) Maximum-Likelihood trees were generated based on 217 aa and 108 aa of partial HEV ORF1 of representative members of the family *Hepeviridae*. Evolutionary analyses were conducted in MEGA version 11.0.11 [64] with 1000 bootstrap reiterations. Le Gascuel (LG) amino acid substitution model with Gamma distribution (G) and Invariable Sites (I) (LG + G + I) was selected based on Find Best-Fit Substitution Model. The trees were rooted using the divergent cutthroat HEV (PisciHEV). Posterior probabilities are denoted at specific branches. Scale bars indicate the estimated number of amino acid substitutions per site. Virus designations include genera, species, genotype, host, and GenBank accession number. Trees were annotated and visualized in FigTree version 1.4.4 (http://tree.bio.ed.ac.uk/ (accessed on 18 March 2022)).

**Figure 6 viruses-14-00905-f006:**
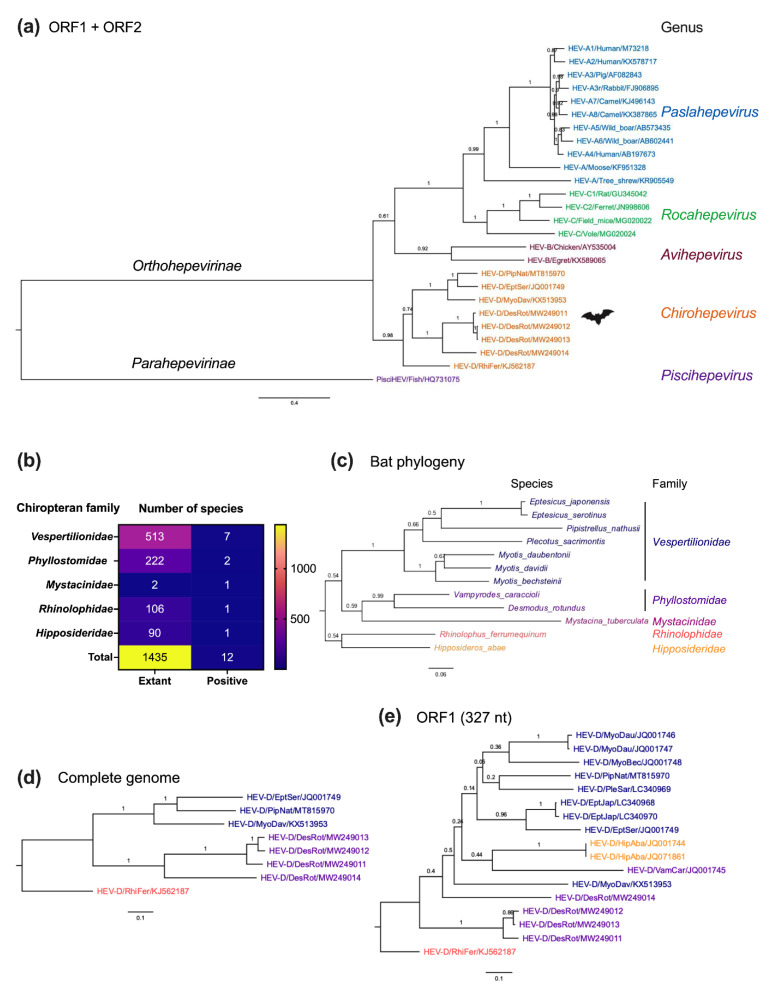
Evolutionary relationships of bat HEV variants and their Chiropteran hosts. (**a**) A Maximum-Likelihood tree was generated based on concatenated HEV ORF1 and ORF2 amino acid sequences of representative members of the family *Hepeviridae*. (**b**) Number of positive sampled Chiropteran species per family. Numbers of taxonomically described Chiropteran species in each family are according to the Bat Species of the World databases (https://batnames.org (accessed on 18 March 2022)). (**c**) A host Maximum-Likelihood tree was generated using complete cytochrome B nucleotide sequences of selective bat species. (**d**,**e**) Maximum-Likelihood trees were generated based on complete genome and 327 nt of partial HEV ORF1 of known members of the *genus Chirohepevirus*. The position of HEV partial ORF1 sequences corresponds to 4282 to 4605 of the HEV prototype Burma strain (GenBank accession no. M73218). Evolutionary analyses were conducted in MEGA version 11.0.11 [64] with 1000 bootstrap reiterations. Le Gascuel (LG) amino acid substitution model with Gamma distribution (G) and different Frequencies (F) (LG + G + F) (**a**), General Time Reversible nucleotide substitution model with Gamma distribution (G), and Invariable Sites (I) (GTR + G + I) (**b**,**c**), and Kimura 2-parameter nucleotide substitution model with Gamma distribution (G) (K2 + G) (**d**) were selected based on the Find Best-Fit Substitution Model. The trees were rooted with the most divergent sequence of each tree. Variants from each of the five distinct HEV genera are highlighted in different colors (**a**). Individual bat family and their corresponding bat HEV strains are highlighted with different colors in (**b**,**c**). Posterior probabilities are denoted at specific branches. Scale bars indicate the estimated number of amino acid or nucleotide substitutions per site. Virus designations include genus, species, genotype, host, and GenBank accession number. GenBank accession numbers of bat species used for analyses: NC_022423 (DesRot), LC361451 (EptJap), MF187951 (EptSer), EU934448 (HipAba), NC_034227 (MyoBec), AY665137 (MyoDau), NC_025568 (MyoDav), AY960981 (MysTub), AJ504446 (PipNat), LC036641 (PleSac), MG921109 (PteVam), AB085731 (RhiFer), FJ154184 (VamCar). Trees were annotated and visualized in FigTree version 1.4.4 (http://tree.bio.ed.ac.uk/ (accessed on 18 March 2022)).

**Table 1 viruses-14-00905-t001:** Current classification of the family *Hepeviridae* ^1^.

Family	Subfamily	Genus	Species	Genotype	Host (Commonly)
*Hepeviridae*	*Orthohepevirinae*	*Paslahepevirus*	*balayani*	HEV-A1	Human
				HEV-A2	Human
				HEV-A3	Human, pig, wild boar, deer, rabbit
				HEV-A4	Human, pig, wild boar
				HEV-A5	Wild boar
				HEV-A6	Wild boar
				HEV-A7	Dromedary camel
				HEV-A8	Bactrian camel
			*alci*		Moose
			Unclassified		Tree shrew (likely)
		*Avihepevirus*	*magniiecur*		Chicken, sparrow
			*egretti*		Little egret
		*Rocahepevirus*	*ratti*	HEV-C1	Rat, house shrew
				HEV-C2	Ferret
				HEV-C3	Field mouse
			*eothenomi*		Vole
			Unclassified		Hamster
		*Chirohepevirus*	*eptesici*		Serotine, myotis bat
			*rhinolophi*		Horseshoe bat
			*desmodi*		Vampire bat
	*Parahepevirinae*	*Piscihepevirus*	*heenan*		Trout

^1^ The table is adapted from the proposed classification of the family *Hepeviridae* according to 2021 ICTV release (https://talk.ictvonline.org/taxonomy/p/taxonomy-history?taxnode_id=202113883 (accessed on 2 April 2022)).

**Table 2 viruses-14-00905-t002:** Detection of HEV-D (genus *Chirohepevirus*) in different bat (order Chiroptera) species.

Family	Species	Common Name ^1^	Sampling Site (Year)	Sample Source	Genomic Sequence (No.)	Reference
*Vespertilionidae*	*Eptesicus serotinus*	Serotine	Germany (2008)	Liver	Complete (1)	[36]
	*Eptesicus japonensis*	Japanese short-tailed bat	Japan (2015)	Feces	Partial (2)	[55]
	*Myotis bechsteinii*	Bechstein’s myotis	Germany (2008)	Feces	Partial (1)	[36]
	*Myotis daubentonii*	Daubenton’s myotis	Germany (2008)	Feces	Partial (2)	[36]
	*Myotis davidii*	David’s myotis	China (2011)	Liver	Complete (1)	[54]
	*Plecotus sacrimontis*	Japanese long-eared bat	Japan (2015)	Feces	Partial (1)	[55]
	*Pipistrellus nathusii*	Nathusius’ pipistrelle	Switzerland (2019)	Feces	Nearly complete (1) ^2^	[59]
*Phyllostomidae*	*Vampyrodes caraccioli*	Great stripe-faced bat	Panama (2011)	Blood	Partial (1)	[36]
	*Desmodus rotundus*	Vampire bat	Peru (2016)	Fecal swab	Complete (4)	[58]
*Mystacinidae*	*Mystacina tuberculata*	New Zealand lesser short-tailed bat	New Zealand (2013)	Feces	Partial (1)	[56]
*Rhinolophidae*	*Rhinolophus ferrumequinum*	Greater horseshoe bat	China (2013)	Fecal swab	Complete (1)	[57]
*Hipposideridae*	*Hipposideros abae*	Aba roundleaf bat	Ghana (2009)	Feces	Partial (2)	[36]

^1^ Common names are given according to the International Union for Conservation of Nature (IUCN) Red List (https://www.iucnredlist.org/ (accessed on 18 March 2022)). ^2^ Viral genome lacks 5′ end.

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
