# Peer review of "Chirohepevirus from Bats: Insights into Hepatitis E Virus Diversity and Evolution"

_viruses, 2022, doi:10.3390/v14050905_

Round 1
Reviewer 1 Report
This review describes the current state of bat hepatitis E virus (HEV) and provides novel insights into genetic diversity, genomic characterization and molecular evolution of bat HEV. In addition, according to the 2021 release of the International Committee on the Taxonomy of Viruses (ICTV), the authors provide a lot of new information regarding bat HEV and other members of the Hepeviridae family to the readers.
The manuscript is well written, and the information gathered is presented in a detailed and comprehensible manner. However, I would recommend the following minor changes:
Line 29: “3000,000” should be corrected to “300,000”.
Line 51: “molecular” should be modified to “molecularly”.
Line 317: “HEV-C-1” should be changed to “HEV-C1”.
Line 396: “FigTree version 1.4.4” needs a relevant reference.
Figures 2-5: The names of bat HEV strains in Figs. 2 and 3 should better be consistent with those in Figs. 4 and 5. For example, how about changing BS7/EptSer in Figs. 2 and 3 to EptSer/JQ001749 as in Figs. 4 and 5?
Author Response
Manuscript ID: viruses-1696906
Wang and Yang “Chirohepevirus from Bats: Insights into Hepatitis E Virus Diversity and Evolution”
Point-by-point response to the Reviewers’ comments
We appreciate the constructive comments by two reviewers to improve the manuscript, and in the revised manuscript, we have addressed each and every comment.
Reviewer #1:
This review describes the current state of bat hepatitis E virus (HEV) and provides novel insights into genetic diversity, genomic characterization and molecular evolution of bat HEV. In addition, according to the 2021 release of the International Committee on the Taxonomy of Viruses (ICTV), the authors provide a lot of new information regarding bat HEV and other members of the Hepeviridae family to the readers.
RESPONSE: We thank the reviewer 1 for his/her thorough review and positive comments.
The manuscript is well written, and the information gathered is presented in a detailed and comprehensible manner. However, I would recommend the following minor changes:
Line 29: “3000,000” should be corrected to “300,000”.
RESPONSE: We have changed “3000,000” to “300,000” (line 29).
Line 51: “molecular” should be modified to “molecularly”.
RESPONSE: We have changed “molecular” to “molecularly” (line 53).
Line 317: “HEV-C-1” should be changed to “HEV-C1”.
RESPONSE: We have changed “HEV-C-1” to “HEV-C1” (line 336).
Line 396: “FigTree version 1.4.4” needs a relevant reference.
RESPONSE: As suggested, we have added a reference for the FigTree software accordingly (lines 417 and 498).
Figures 2-5: The names of bat HEV strains in Figs. 2 and 3 should better be consistent with those in Figs. 4 and 5. For example, how about changing BS7/EptSer in Figs. 2 and 3 to EptSer/JQ001749 as in Figs. 4 and 5?
RESPONSE: Thanks for this valuable comment. Since the names of bat HEV strains (e.g., BS7/EptSer) have occurred multiple times in our manuscript (please see lines 170–176, 237–243, and 296), we tend to keep this useful information in Figure 2 and Figure 3. Instead, we have added the GenBank accession numbers of each of the eight bat HEV strains in the figure legends of Figure 2 and Figure 3 (lines 288–290 and 350–353), and we believe this can improve the consistency of the names of bat HEV strains in the figures.
Reviewer #2:
I have read with interest the review on Bat HEV. Authors have gone over the classification/ Taxonomy of HEV as of ICTV taxonomy report of 2021. In this report, HEV family has been divided in to 2 subfamilies, 5 genera and 10 species. All subfamilies, genera and species have been given biologic names which need to be followed up in future by all involved in HEV research. Bat HEV under the title of Chirohepevirus with 3 species is the focus of review in this paper. Authors have gone to details of the bat family and given classification of this family with HEV viruses. A global map of Bat HEV is enclosed for this review. Authors have correctly pointed to the fact that no human infections have been reported as of today, however, taking the heterogeneity of the agent we need to be open to this possibility in future.
RESPONSE: We thank the reviewer 2 for his/her constructive comments to improve our manuscript.
While going through the review I thought following points which could be edited or added to review.
- A tabulated format of ICTV 2021 classification would be useful to reduce unnecessary description of this report which may be confusing to readers.
RESPONSE: As suggested by the reviewer, we have added a tabulated format of HEV classification according to the 2021 ICTV release in the revised manuscript (Table 1).
- Authors have given details description of genome of Bat HEV and how it differs from another genomes. A sketch of major genomes with ORF’s may be useful to make this point with clarity. Comparisons of mammalian, avian, bat, rat and fish genomes may be adequate.
RESPONSE: Thanks for the reviewer’s valuable suggestion, we have added a sketch to compare the genomic organization of representative HEV strains from different HEV genera, including human HEV-A, avian HEV-B, rat HEV-C, bat HEV-D, and fish PisciHEV (Figure 3).
- Authors mentioned that HEV A1 has ORF4 which is important in biology and pathogenesis of this epidemic strain of the virus. Need better description of this ORF. Also, rat HEV also has ORF4 which has different significance. Authors need to make description of these 2 ORF4’s exact and define their significance.
RESPONSE: We have added more descriptions of the significance of these two putative ORFs in our revised manuscript accordingly (lines 329–339).
- Authors mentioned (end of the first para) that molecular mechanism in pathogenies of HEV under different situation (pregnancy, chronic HEV and neurologic HEV) is unknown. However, for HEV in pregnancy there is enough evidence that ORF4 of HEV A1 is fundamentally important in the etiopathogenetic mechanism of increased incidence and severity in pregnant women (Viruses2021, 13(7), 1329; https://doi.org/10.3390/v13071329). This needs to be clarified.
RESPONSE: According to the reviewer’s comment, we have added a statement that “The altered hormone levels and immunologic responses may contribute to the severity of liver diseases in pregnancy” and cited the excellent review article by Khuroo MS, Viruses 2021 accordingly (lines 35–36). In addition, we have softened our statement and rephrased “unknown” to “poorly understood”. However, we are very prudent with the correlation between the putative HEV-A1 ORF4 and severe liver diseases in pregnant women since there is insufficient experimental data to confirm this hypothesis.

Reviewer 2 Report
I have read with interest the review on Bat HEV. Authors have gone over the classification/ Taxonomy of HEV as of ICTV taxonomy report of 2021. In this report, HEV family has been divided in to 2 subfamilies, 5 genera and 10 species. All subfamilies, genera and species have been given biologic names which need to be followed up in future by all involved in HEV research. Bat HEV under the title of Chirohepevirus with 3 species is the focus of review in this paper. Authors have gone to details of the bat family and given classification of this family with HEV viruses. A global map of Bat HEV is enclosed for this review. Authors have correctly pointed to the fact that no human infections have been reported as of today, however, taking the heterogeneity of the agent we need to be open to this possibility in future.
While going though the review I thought following points which could be edited or added to review.
- A tabulated format of ICTV 2021 classification would be useful to reduce unnecessary description of this report which may be confusing to readers.
- Authors have given details description of genome of Bat HEV and how it differs from another genomes. A sketch of major genomes with ORF’s may be useful to make this point with clarity. Comparisons of mammalian, avian, bat, rat and fish genomes may be adequate.
- Authors mentioned that HEV A1 has ORF4 which is important in biology and pathogenesis of this epidemic strain of the virus. Need better description of this ORF. Also, rat HEV also has ORF4 which has different significance. Authors need to make description of these 2 ORF4’s exact and define their significance.
- Authors mentioned (end of the first para) that molecular mechanism in pathogenies of HEV under different situation (pregnancy, chronic HEV and neurologic HEV) is unknown. However, for HEV in pregnancy there is enough evidence that ORF4 of HEV A1 is fundamentally important in the etiopathogenetic mechanism of increased incidence and severity in pregnant women (Viruses2021, 13(7), 1329; https://doi.org/10.3390/v13071329). This needs to be clarified.
Author Response

(The authors gave the same response as above.)
